# The Evolving Landscape of Functional Models of Autism Spectrum Disorder

**DOI:** 10.3390/cells14120908

**Published:** 2025-06-16

**Authors:** Jai Ranjan, Aniket Bhattacharya

**Affiliations:** 1Department of Microbiology, All India Institute of Medical Sciences, Bathinda 151001, India; jairanjan5@gmail.com; 2Department of Neuroscience and Cell Biology, Child Health Institute of New Jersey, Rutgers Robert Wood Johnson Medical School, New Brunswick, NJ 08901, USA

**Keywords:** ASD, autism, mouse model, SHANK3, CHD8, MECP2, FMR1, iPSC, organoid, assembloid

## Abstract

Autism spectrum disorder (ASD) is a neurodevelopmental disorder affecting 1–3% of the population globally. Owing to its multifactorial origin, complex genetics, and heterogeneity in clinical phenotypes, it is difficult to faithfully model ASD. In essence, ASD is an umbrella term for a group of individually rare disorders, each risk gene accounting for <1% of cases, threaded by a set of overlapping behavioral or molecular phenotypes. Validated behavioral tests are considered a gold standard for ASD diagnosis, and several animal models (rodents, pigs, and non-human primates) have traditionally been used to study its molecular basis. These models recapitulate the human phenotype to a varying degree and have been indispensable to preclinical research, but they cannot be used to study human-specific features such as protracted neuronal maturation and cell-intrinsic attributes, posing serious limitations to translatability. Human stem cell-based models, both as monolayer 2D cultures and 3D organoids and assembloids, can circumvent these limitations. Generated from a patient’s own reprogrammed cells, these can be used for testing therapeutic interventions that are more condition and patient relevant, targeting developmental windows where the intervention would be most effective. We discuss some of these advancements by comparing traditional and recent models of ASD.

## 1. Introduction

Neurodevelopmental and neuropsychiatric disorders (NDDs and NPDs) impose an ever-increasing burden on global health today [1,2]. Two of the key reasons behind these high numbers are improvement in diagnostic reach and availability of relatively cheap genetic tests [3]. Children are now routinely screened for NDDs at a behaviorally relevant developmental stage and referred to genetic tests such as clinical exome sequencing to detect known genetic variants [4].

Autism spectrum disorder (ASD), characterized by restricted, repetitive behavior and interests as well as impairment in social communication and adaptive difficulties, is an NDD affecting around 1–3% of individuals globally [2,5,6]. According to the Diagnostic and Statistical Manual of Mental Disorders, 5th Edition (DSM-V), ASD encompasses several NDDs such as autism, childhood disintegrative disorder, Rett syndrome, pervasive developmental disorder, etc., arranged hierarchically based on severity [7,8]. Notwithstanding improved diagnosis, there are several roadblocks in research towards finding treatments that can improve the quality of life of individuals with ASD [9]. First, ASD is multifactorial with an enormous genetic heterogeneity [10]. The genetic risk factors include rare inherited as well as *de novo* variants, both point mutations and copy number variants (CNVs), with each of them individually explaining less than one percent of ASD genetic risk [11]. This genetic burden is also exacerbated by non-genetic factors such as paternal age at conception and hormones that a developing brain is exposed to in the womb (maternal microenvironment). For instance, males outnumber females at a ratio of 4:1 for ASD, which is explained by two competing theories suggesting that females have a higher threshold of tolerance for genetic burden linked to ASD (the female protective effect) [12], or males are more susceptible (male risk or “extreme male brain”) [10]. This can also be modulated by epigenetic factors, which, in turn, are responsive to the presence of agents such as alcohol, nicotine, hypoxia, valproic acid, glucocorticoids, and prenatal infection in the maternal microenvironment [13,14,15,16]. Second, this genetic complexity is mirrored in the heterogeneity of clinical phenotypes ranging from autistic individuals with low support needs who can function independently with little external accommodations to those with profound ASD that require constant care. This is also complicated by the presence of comorbidities like intellectual disability (ID), attention deficit hyperactivity disorder (ADHD), seizure, epilepsy, aggression, anxiety, depression, sleep anomalies, microcephaly, regression, speech deficits, gastrointestinal disorders, etc. [3,17,18]. Third, ASD is diagnosed through behavioral deficits using validated tests such as the Childhood Autism Rating Scale (CARS), Autism Diagnostic Interview-Revised (ADI-R), and Autism Diagnostic Observation Schedule (ADOS), administered to children who are 18–24 months and older [19,20,21,22,23,24]. Since the human brain continues to develop through infancy [25], an earlier diagnosis can vastly improve therapeutic efficacy by targeting a more appropriate window of brain development. Lastly, it is difficult to develop ASD models that recapitulate the disorder in its entirety, mainly because of its clinical and genetic heterogeneity [26,27,28,29,30,31,32,33]. Animals such as rats and mice have traditionally been used to model ASD phenotypes like hyperactivity, aggression, repetitive behaviors, deficits in memory and sociability (in terms of behavioral modalities), as well as microcephaly and myelination deficits (in terms of brain morphology). While these models have contributed to a wealth of studies that have significantly expanded our understanding of ASD, animal models present with some inherent limitations to translatability, such as their inability to recapitulate human-specific transcriptional paradigms and the protracted development of the human brain [25,34,35]. Over the last decade, human pluripotent stem cell (hPSC)-derived 2D models such as induced neurons (iNs) as well as 3D models (organoids, assembloids) have started to bridge this gap [36,37]. Patient-derived iNs offer an opportunity to study deficits in neuronal maturation *in vitro*, and because hPSCs can be expanded in culture, this model offers a seemingly infinite source of genetic background-matched cells on which several drugs can be tested in tandem [36]. Such an approach embodies the essence of personalized medicine while ensuring a quick turnaround to target critical developmental windows for treatment that is most appropriately designed for each individual [25,36,38].

Here we review the genetic and environmental risk factors for ASD and how those are modeled in animals as well as hPSC-derived 2D and 3D models. We highlight the strengths and weaknesses of each model and advocate for orthogonal validation across different model systems while also hedging claims to what a certain ASD model can unequivocally establish.

## 2. Genetic Risk Factors

The involvement of genes in the etiology of ASD has now been firmly established [39,40,41]. While heritability estimates in studies comparing monozygotic and dizygotic twins have previously suggested a strong genetic component [42,43,44], recent large-scale, genome-wide studies have come up with a list of high-confidence ASD risk genes [45,46,47]. The advent of newer technologies like genome-wide association studies (GWAS), whole genome sequencing (WGS), whole exome sequencing (WES), and clinical exome sequencing has led to the discovery of a trove of more than 1100 ASD risk genes at varying levels of confidence (https://gene.sfari.org/database/gene-scoring/ accessed on 12 June 2025) [45,46,47,48]. While rare variants of large effect size often follow a Mendelian mode of inheritance (autosomal dominant, recessive, X-linked), common variants cumulatively add to polygenic risk [40,49]. The variants can be both point mutations, such as single nucleotide variants (SNVs), missense or protein truncating, as well as structural variants present in variable copies due to deletion or duplication of certain genomic loci [50,51]. Although most of these variants are exonic/coding, some non-coding ASD risk alleles have also been reported [52].

### 2.1. Inherited Genetic Variants

Monogenic factors of large effect size often contribute to syndromic forms of ASD (i.e., ASD is present as a part of the wider clinical presentation), such as loss of function (LoF) of *FMR1* in Fragile X syndrome, *TSC1* and *TSC2* in tuberous sclerosis complex, and *MECP2* in Rett syndrome [53,54,55,56]. On the other hand, disorders such as Timothy syndrome and *PTEN* LoF present ASD as an auxiliary phenotype alongside cardiovascular defects and cancer, respectively [57,58,59]. Studies on syndromic autism and ID have been informative in establishing the gene dosage sensitivity in the developing brain [54,60,61].

Inherited recessive variants explain only a fraction (2–19%) of ASD risk. Unique demographic features such as consanguinity and endogamy are associated with the increased load of rare, recessive variants in ASD risk genes [62,63,64,65]. Genetic studies on multigenerational consanguineous families have been seminal in the discovery of ASD risk genes through classical logarithm of the odds (LOD) score analysis [39,62,63,66]. Rare LoF variants in high-confidence ASD risk genes like *SHANK3* and *MECP2* are highly penetrant and have also been modeled in rodents to decipher their function in brain development. Individual common variants do not usually predispose individuals to a higher risk of developing ASD, but cumulatively common SNPs explain 17–52% of ASD risk [28,40,67]. Polygenic risk scores are calculated by adding the risk of all SNPs. Each SNP is assigned a specific weightage by utilizing algorithms like PRSice-2 (v2.3.5), which can inform ASD risk due to common variants [68,69]. This is particularly helpful in inferring oligogenic and polygenic effects where the presence of two or more genetic variants (hits) is a prerequisite for ASD to occur.

### 2.2. De Novo Variants (DNV)

Approximately 85% of ASD cases are idiopathic (without any known genetic etiology), a large fraction of which can be attributed to DNVs [48,70]. DNVs are novel mutations that arise in the ASD patients and are absent in their parents and siblings. The expected rate of DNV is estimated to be ~2 × 10^8^/base/generation [71]. WES studies on simplex families with trios (parents + affected child) and quads (trio + unaffected sibling) have been instrumental in the discovery of DNVs in ASD risk genes, particularly among sporadic cases [29,71,72,73]. DNVs can be missense as well as likely gene disrupting (LGD). The presence of multiple independent DNVs in the same gene across unrelated individuals with ASD provides stronger evidence for their involvement in the disorder [29]. CNVs with coding DNVs explain 30% of simplex families (25% boys and 45% girls), with the females sharing more LGD mutations with lower IQ males than those with high IQ. There is an increased prevalence of gene-disrupting mutations in ASD cases compared to controls, majorly in brain-expressed genes, with a heavy bias (4:1) towards paternal age [74,75]. Early transcriptional regulators, but not Mendelian ID risk genes, are enriched for rare ASD-related DNVs [76].

### 2.3. Convergent Signaling Modules

Despite this immense genetic heterogeneity, ASD risk genes converge on shared pathways such as regulation of protein synthesis (especially synapse-related proteins), transcriptional regulation and chromatin remodeling, and regulation of cell proliferation in the developing brain [29,45,75,77] (reviewed in [39,40]). Both cis and trans regulatory targets of ASD genes are enriched for ASD and other NDD risk genes [54,78,79]. Studies targeting such convergent modules have been able to posit genetic findings in a mechanistic perspective to aid drug discovery [80].

For example, *SHANK* is one of the major gene families linked to ASD. These genes encode postsynaptic proteins that regulate synaptic transmission [81]. *SHANK1* and *SHANK3* play a role in postsynaptic density maintenance, while *SHANK2* is primarily involved in synaptic signaling [82,83]. LoF of either gene can thus lead to defective synaptic function and cause ASD. The developing brain is, in fact, so sensitive to the expression of these key proteins that their haploinsufficiency (loss of one allele) is a leading genetic cause of ASD (Table 1). Similarly, LoF of *ANK3* encoding for ankyrin-G, a scaffold protein that links the cytoskeleton to membrane proteins, leads to defective neurogenesis [84,85]. The male-specific lethal (MSL) complex, present on the X chromosome, leads to the production of at least five proteins, viz., MSL1, MSL2, MSL3, MOF, and MLE [86]; DNVs in *MSL2* have been implicated in ASD [87]. A representative list of some high-confidence ASD risk genes has been provided in Table 1. Most of these genes are crucial to neuronal biology and expressed during brain development.

### 2.4. Variants of Uncertain Significance (VUS)

Fueled by the growth in WES, an increasing number of clinical variants are now being discovered, but a bulk of them are subsequently classified as VUSs. These are mostly missense variants with unknown effect on gene function, often with conflicting annotation of pathogenicity. VUSs represent one of the major bottlenecks in using patient genetics to inform clinical decisions, such as determining timelines when an intervention would be most effective or choosing from an array of available drug candidates the one that would be most likely to work in a particular patient [40,88]. Since ASD risk genes display convergence at the molecular level, functional assays interrogating variant effect on these nodes, such as CREB signaling, can be informative in segregating pathogenic variants from benign ones and help in VUS classification [17,89,90,91].

**Table 1 cells-14-00908-t001:** A representative panel of some high-confidence ASD risk genes along with their phenotypes in mutant animal models and associated NDDs.

Genes	Protein Function/Role in ASD	Chromosomal Location	Mouse Mutation	Mutant Phenotype	Associated NDDs	Limitations/Strengths	Reference
*ANK3*	Scaffold protein, links membrane protein to cytoskeleton	10q21.2	Heterozygous KO	Increased anxiety, smaller cortex	Bipolar disorder, Schizophrenia	Lacked exon 37 genomic sequence, driver expressed Cre with enhancer which is less efficient in PNS	[85,92,93]
*CHD8*	Chromatin remodeling	14q11.2	Heterozygous KO	Altered social behavior, increased anxiety, repetitive behavior	Overgrowth with macrocephaly, ID	No conclusive evidence of *CHD8* haploinsufficiency and abnormal REST activation	[94,95,96,97]
*CNTNAP2*	Cell adhesion molecule responsible for interaction between glia and neurons	7q35-q36.1	Homozygous KO	Mitochondrial dysregulation, axonal impairment, synaptic vesicle transport disruption	Cortical dysplasia, focal epilepsy, ID, language impairment, Tourette syndrome	Longer cue duration bisection tasks not employed; rearing influences not considered	[98,99,100,101,102,103]
*FMR1*	Synaptic mRNA translation; leading genetic cause of ASD	Xq27.3	Hemizygous and Homozygous KO	Learning deficits, hyperactivity, dendritic spine maturation defects	Fragile X-associated tremor/ataxia syndrome (FXTAS), ID	Confirmation of behaviorally induced modulation of neuroplasticity by showing causal inverse relationship between behavior and neuroplasticity not performed; High level FMRP overexpression in *Fmr1* KO mice cause aberrant behavior	[56,104,105,106,107]
*GRIN2B*	NMDA receptor ion channel subunit: *de novo* mutations result in neuronal circuit alterations	12p13.1	Heterozygous KO, conditional KO	More spontaneous spikes and wave discharges	Epilepsy, ADHD, Schizophrenia, ID, developmental delay	Failure to suckle and death in mouse models; ASD; hypersensitivity due to supraspinal mechanisms	[108,109,110,111]
*MECP2*	Binds to methylated cytosines	Xq28	Hemizygous and Homozygous KO	Battery of neurological phenotypes, hindlimb clasping	Rett syndrome, ASD, Epilepsy, regression		[55,112,113,114,115,116]
*MSL2*	Biallelic expression of dosage sensitive genes, histone H4 acetylation	3q22.3	Homozygous KO	Heterogenous phenotype and perinatal lethality	Karayol-Borroto-Haghshenas NDD syndrome	Multiple cell type leads to heterogeneity	[61,117]
*NLGN4X*	Post-synaptic cell adhesion molecule, binds neurexin	Xp22.32-p22.31	Hemizygous and Homozygous KO	Reduced excitation: inhibition ratio, reduced social interaction, vocalization	Tourette syndrome, Fragile X syndrome	Network stimulation with 3D multi electrode array used to determine function of *Nlgn4* at cortical column	[118,119,120,121]
*NRXN1*	Pre-synaptic cell adhesion molecule, binds neuroligin	2p16.3	Homozygous KO	Affected social novelty preference, increased aggression in males	ADHD, Schizophrenia	Nrxn1α KO mice engineered from SV129 mice as these can be targeted by homologous recombination.	[122,123]
*PTCHD1*	Transmembrane protein, Sonic hedgehog signaling	Xp22.11	Hemizygous and Homozygous KO	Cognitive deficits, synaptic gene expression changes, excitatory synaptic dysfunction	ID	No localization or fluorescence in transfected cells with N terminal GFP tag	[124,125,126,127]
*PTEN*	Synaptic inhibition and excitation imbalance	10q23.31	Heterozygous KO, conditional KO	Synaptic alterations, hyperactive hippocampal mTOR signaling, microglia activation	Macrocephaly, epilepsy	Only SSC region analyzed	[58,59,128,129,130]
*RELN*	Cell positioning during brain development; loss results in neuronal dysplasia	7q22.1	Homozygous KO	“Reeler” mice, lamination defects in brain, deficiency in neurogenesis	Epilepsy, Schizophrenia, Bipolar disorder, Lissencephaly with cerebellar hypoplasia	Functional impairment more in *Reln* deficient mice	[131,132,133,134]
*SCN2A*	De novo mutations impair voltage-gated sodium channels and affects dendritic excitability	2q24.3	Homozygous KO, Heterozygous KO, conditional KO	Delayed spatial learning, disrupted nesting, mating, anxiety	Epilepsy, movement disorder	*Scn2a* KO mice showed increased anxiety and little or no mating and nesting, suggesting behavioral abnormalities	[135,136,137,138,139]
*SHANK* family	Encodes post-synaptic scaffold proteins, responsible for ~1% of ASD cases	*SHANK3* (22q13.33)	Homozygous KO	Abnormal social interaction	Phelan–McDermid syndrome, Schizophrenia, ID, ASD, epilepsy, developmental delay	Mutant Shank3B affect social interaction	[81,140,141,142,143]
*UBE3A*	E3 ubiquitin ligase, alteration of synaptic function	15q11.2	*Ube3a* ^m-/p+^	Motor dysfunction, inducible seizures, impaired LTP	Angelman syndrome, ID, ataxia	Importance of nuclear UBE3A established	[144,145,146,147]

## 3. Environmental Risk Factors

More than half of the ASD cases cannot be explained by genetics, suggesting environmental factors that alone or in conjunction with genes predispose individuals to ASD risk [16]. These can broadly be classified into external and internal factors, based on whether it is an external agent such as a drug or toxin or an internal body state such as maternal stress or diabetes. The effect of these environmental (i.e., non-genetic) agents is maximal *in utero*. These can particularly affect the developing brain during the critical window of neurogenesis, from about the 40th day to the 125th day of gestation, i.e., the first to early second trimester in humans [148] and their effects may be modulated by altering the maternal microenvironment.

### 3.1. External Factors

Use of antiepileptogenic drugs like valproic acid (VPA) by pregnant women has long been known as a strong prenatal risk factor for ASD and ID, which has since been validated in multiple model systems [15,149,150,151,152,153]. Histone acetylation has been associated with ASD in recent studies because VPA can lead to histone deacetylase inhibition [149,154]. Nitric oxide (NO) has also been suggested as an ASD risk factor, with studies in ASD mouse models reporting an increased NO level. This is further supported by a phenotypic rescue upon treatment with a NO synthase inhibitor in *Cntnap2* and *Shank3* mouse models [155]. Exposure to air pollutants (particulate matter) and pesticides has also been correlated with ASD due to their neurotoxicity [156,157]. Similarly, environmental exposure to chemicals such as inorganic mercury, thiomersal, and heavy metals was also found to influence ASD [158,159]. Smoking and alcohol consumption in pregnancy have also been designated as ASD risk factors. Selective serotonin reuptake inhibitors (SSRIs) and certain antibiotics during pregnancy also contribute to ASD risk, especially if administered during critical windows of brain development [160,161,162,163].

### 3.2. Internal Factors

Internal factors such as infections during pregnancy, gestational diabetes, and maternal obesity could result in inflammation, mitochondrial dysfunction, and alteration in gut microbiome [164]. A disrupted gut–brain axis due to altered gut microbiota could lead to ASD, as brain development is influenced by maternal gut microbiome [165]. Certain maternal blood metabolites, such as trimethylamine-N-oxide (TMAO), imidazole propionate, etc., have been shown to lead to an increase in axon numbers due to thalamocortical axonogenesis in mice [166]. A decrease in their levels due to an altered gut microbiota can therefore lead to ASD. Other neonatal factors that have been implicated in the development of ASD are congenital heart disease, low birth weight, macrosomia, male sex, as well as APGAR score less than 7 at 1 min [167].

## 4. Animal Models of ASD

Autism was first defined as a discrete category of NPD by Leo Kanner in 1943 using a set of behavioral phenotypes that included stereotypy, obsession for a set routine, echolalia, and deficits in social communication, such as altered gaze [168]. To this day, ASD is diagnosed through a standardized battery of behavioral tasks and questions [20,21,22,23,24]. Consequently, preclinical animal models that can reliably simulate ASD-relevant human behaviors have been seminal to research [169,170,171,172,173,174]. A perfect animal model of ASD is untenable because autism, especially facets of it that arise at the interface of developmental gene expression programs and neuronal circuits, is human specific [175]. Yet, reductive animal models of ASD have been created by introducing syntenic mutations (genetic models) as well as *in utero* exposure to metabolites like VPA (environmental models) [176,177] (Figure 1). When choosing a particular model, researchers should ensure it meets both construct validity (i.e., it faithfully recapitulates the underlying genetic or environmental cause, such as a patient mutation) as well as face validity (close similarity to the human clinical phenotype) [171,175]. The essential criteria for designing ASD mouse studies have been proposed by collating information from mouse models of high-confidence ASD risk genes [175]. Common behavioral tests employed for the evaluation of ASD-like behaviors in mice include the three-chambered assay for sociability, ultrasonic vocalization for communication, marble burying for stereotypy, and the T-maze or Morris water maze for spatial memory as well as cognitive flexibility [177,178]. Each model comes with its pros and cons; inferences need to be hedged based on evolutionary closeness and naturalistic tendencies, which generally inform the extent of translatability to humans. For instance, zebrafish embryos are transparent, so defects in brain development can be easily visualized provided the region is conserved [179,180,181]. Mice are neophiliac and social in nature, so ASD mouse models can be tested for their aversion to novelty and sociability. Prairie voles have also been used to study many aspects of social behavior and parental care because they form long-term monogamous pairs [182,183]. A variety of animals have been used to model ASD phenotypes [169,176,184,185,186,187,188,189,190,191,192,193], of which mice, rats, pigs, and non-human primates are discussed below.

### 4.1. Murine Models of ASD

Rodent models of ASD are popular due to their shorter generation time and limited requirement of housing space. Additionally, there is a wealth of genetic tools and resources available for mice, including transgenic reporter lines, CRE (recombinase) driver lines to allow spatiotemporally restricted gene expression, inducible lines that express certain genes only when drugs such as tamoxifen or doxycycline are administered, etc. [175].

Syndromic forms of ASD, such as Phelan–McDermid syndrome caused by *SHANK3* haploinsufficiency, have been extensively studied (reviewed in [194]). *Shank3* knockout (KO) mice respond differently to chronic social isolation than their wild-type (WT) littermates [195]. WT mice exhibit enhanced social interaction, but social novelty preference and self-grooming were restored in *Shank3* KO mice. Dysregulations in the social circuit were found in three different KO mice (*Shank3*, *Fmr1,* and *Oprm1*) subjected to early social isolation, highlighting the developmental role of these ASD risk genes [196]. This effect is not just restricted to neurons, as evidenced in the study of Fischer et al., using InsG3680 mice to evaluate the role of *Shank3* in postsynaptic oligodendrocyte and its effect on myelination [83]. Imbalance in GABA (a neurotransmitter) and decreased oxytocin level are hallmarks of ASD. The neuropeptide oxytocin (OXT) and its paralog vasopressin regulate social interactions to influence behavior. These are synthesized in the paraventricular and supraoptic nuclei (PVN and SON, respectively) of the hypothalamus. Pathogenic variants in the *OXT* gene as well as its receptor (*OXTR*) have been reported in individuals with ASD. Three genetic mouse models, *Shank3*, *Fmr1,* and *Oprm1* KO mice subjected to early chronic social isolation, were evaluated for *Egr1*, *Foxp1*, *Homer1a*, *Oxt,* and *Oxtr* expression as molecular markers for immediate early gene families. This study provided proof-of-concept for stratification of mouse models. In three-chamber tests, *Fmr1* KO mice demonstrated significant sociability deficits, *Shank3* KO mice showed decreased social novelty preferences, and *Oprm1* KO mice showed a lack of mate preference, while in the reciprocal social interaction tests, only *Fmr1* KO mice had decreased interaction with stranger mice. In terms of gene expression, *Fmr1* KO mice showed a decrease in expression of all four mRNAs in PVN; *Oxt* and *Avpr1a* were downregulated in *Oprm1* KO, and *Shank3* KO mice exhibited downregulation in the prefrontal cortex and nucleus accumbens [196].

Haploinsufficiency of certain ASD risk genes, such as *SYNGAP1*, can also lead to NDDs. SynGAP inhibits AMPA receptor insertion in the postsynaptic membrane by inhibiting the RAS pathway and is majorly expressed in the striatum in mice during the first few postnatal weeks. The striatum has striatal projection neurons (SPNs) with dopamine receptors whose activation is thought to suppress locomotor activity. Interestingly, *Syngap1* mRNA is expressed in SPNs, and a reduction in its level affected the motor performance of mice, signifying its role in ASD-like phenotypes [197]. Similarly, decreased reactivity in response to painful stimuli via S1PR1 (associated with sphingolipid metabolism) has been studied in the peripheral nervous system of BTBR T+tf/J mice. It involves KCNQ/M channels in the dorsal root ganglion, is mediated through the MAPK and cAMP/PKA pathways, and leads to a smaller action potential [198]. Tuberous sclerosis complex (TSC) is associated with ASD; mutations in *TSC1*/*TSC2* can result in inhibition of the mechanistic target of rapamycin (mTOR) pathway. ASD-like behavior is observed among mice with *Tsc1* deletion. In VPA-treated mice, increased mTOR proteins are found in various brain regions, which correlate with decreased autophagy. For example, upon exposure to VPA, a diminished catecholamine level is observed in the striatum, which is associated with increased mTOR signaling [199].

Mutations in several other neuronal genes have been found to be associated with ASD. Transcription factor 4 (TCF4), a helix-loop-helix transcription factor located at 18q12.2, is expressed in GABAergic interneurons [200,201]. *TCF4* mutant neuronal progenitors exhibit reduced proliferation and neuronal differentiation defects [202]. Another risk gene, zinc-finger E-box-binding homeobox 2 (*ZEB2*), mainly plays a role in organogenesis and formation of the neural crest. More than 280 *ZEB2* variants have been found in patients with Mowat–Wilson syndrome (MWS), in which one of the most common presentations is ID, with variable presentation of ASD [203]. A duplication syndrome of the X-linked gene methyl CpG-binding protein 2 (*MECP2*) is also associated with ASD [204]. LoF mutations in this gene can weaken neuronal connections and synaptic plasticity [205]. Xu et al. have shown that the development of gray matter volume, mainly in the thalamus and hippocampus, is altered in *MECP2* duplication rat models [206]. The chromodomain helicase DNA binding 8 (*CHD8*) gene regulates chromatin remodeling and has a role in the development of neurons and myelination [207]. Niu et al. showed that it plays a major role in hippocampal neurogenesis in mice. *Chd8* heterozygous mice exhibit anxiety-like behavior and a reduction in neurogenesis, with fewer neurons in the hippocampal region [208]. However, CHD8-deficient mice have been shown to have an expansion of amygdala and hippocampal regions, which may be due to the abnormal activation of the transcription factor REST [209].

ASD pathology also involves other brain cell types such as microglia. VPA-exposed mice as well as *Cntap2* KO mice, when treated with PLX5622, a small molecule inhibitor of colony stimulating factor 1-receptor that depletes microglia, exhibit effective amelioration in ASD-like phenotypes. This is due to the restoration of microglia morphology in hippocampal CA1, striatum, and somatosensory cortex [210].

Besides single-gene models, structural variants linked to ASD have also been created in mice [51]. 16p11.2 deletion is a major genetic driver of multiple NDDs. Mice carrying 16p11.2 deletion are smaller in size, weigh less, and are more severely affected compared to those carrying reciprocal duplication [211,212]. The 17p11.2 deletion, also associated with ASD, can result in craniofacial abnormalities. The 7q11.23 deletion phenotype is driven by the *GTF2I* gene. General transcription factor I (GTF2I) inhibits transcription by binding to histone lysine demethylase 1. 7q11.23 heterozygous duplication is also observed among ASD patients. *Gtf2i* duplication in mice also leads to ASD-like features, suggesting a narrow homeostatic level of this protein is required for physiological brain function [213].

Epigenetic modifications have also been shown to be associated with ASD. Non-coding RNAs, DNA methylation, and histone modifications can predispose an individual to ASD [214,215]. Histone modifications can result in anomalies with synaptic excitation and inhibition [215]. Lysine-specific demethylase 6B (*KDM6B*) is an H3K27me3 demethylase that has been shown to influence ASD risk. *KDM6B* knockout shows defects in synaptic transmission and decreased plasticity [216]. Broad enhancer-like chromatin domains (BELD) contain a set of genes that can result in histone modifications and are also associated with increased RNA polymerase II transcription. These regions are rich in ASD risk genes and are downregulated in ASD cases [217]. Recently, non-coding variants have also been found to be associated with ASD. Shin et al. evaluated human accelerated regions (HARs) and neural VISTA enhancers (VEs) and showed that these regions also contribute to ASD risk [218].

Issues with coordinated brain activity can also be found in ASD. Resting-state fMRI (rs-fMRI) has recently shown promise in detecting aberrant connectivity in ASD. In the Autism Mouse Connectome (AMC) study, multiple ASD mutants were studied. rsfMRI of 16 cohorts with 350 mice from Italy and Switzerland was utilized. Their findings reveal that there are defects in functional connectivity with a significant level of heterogeneity in ASD mouse models. Additionally, there are certain commonalities that are observed between these circuits, which result in some “familial” or “cluster” pattern of ASD [219]. The various challenges of rodent fMRI, such as its prohibitive cost, unclear human–rodent homology, no clear consensus regarding rodent data processing, and availability of only limited high-powered studies, currently impede full usage of this modality [220,221].

### 4.2. Porcine Models of ASD

While multiple modules of ASD-related human behavior have successfully been modeled in rodents, they have limited similarity with humans, especially in terms of neuroanatomy and physiology. Larger mammalian models such as pigs exhibit a greater neuroanatomical similarity with humans, including a gyrified cortex, which the mice lack. Transgenic pig models can assist in deciphering gene-specific ASD pathology and their neurodevelopmental effects. Porcine models have been established for certain syndromic forms of ASD, like Timothy syndrome 1 (*CACNA1C*^Gly406Arg/WT^), although brain phenotypes have not yet been reported [57,222]. Porcine KOs of single NPD genes like *MIR137* are better representative of the human phenotype, especially changes in neuronal transcriptome, and are more similar to human iPSCs than mouse models at the molecular level [223].

Environmental causes of ASD, such as valproic acid, have also been modeled in pigs. Bama miniature pigs embryonically exposed to VPA exhibit heightened anxiety (less time in the center in open field tests), increased aggression to an intruder, increased latency during novel object exploration and spatial memory deficits, along with concomitant cortical malformation, decreased dendritic complexity, and spine maturation. Altered social interaction in the modified three-chamber sociability test was indicated by shorter latency of first sniffing of same-sex unfamiliar pigs. This was also corroborated with significant alterations in dopamine signaling genes (*DDC*, dopamine receptors, *KATNAL2*, *ADORA2A*, and *SCNA4*) in the brain [150].

### 4.3. Non-Human Primate Models of ASD

Although rodent models have greatly contributed to our mechanistic understanding of ASD neurobiology, their evolutionary separation from humans, neuroanatomical differences in key brain regions linked to ASD-relevant behavior, and lack of certain molecular correlates raise concerns about their translatability to patients. Non-human primates (NHPs) such as old world monkeys (*Macaca fascicularis* and *Macaca mulatta*) as well as new world monkeys (marmosets; *Callithrix jacchus*) circumvent a lot of these problems and present ASD animal models with increased translational value [224] (reviewed in [189,225,226]).

NHP ASD models resemble humans in both structural and molecular features of brain organization. For example, rodents have a small prefrontal cortex (PFC) and lack the dorsolateral PFC altogether. The neocortex—a cerebral zone linked to multiple social behaviors—constitutes a much smaller proportion of the rodent brain (~30% in rats) compared to primates (70–80%). These differences are further accentuated by divergence in gene expression pattern and regulation as well as lack of certain neuronal cell types in the rodent brain [227,228]. For instance, the highest expression of *Shank3* is in the PFC in macaques, while in mice it is in the striatum. In addition to their neuroanatomical similarity, NHP ASD models also recapitulate the bulk of the rich repertoire of natural human behavior involving facial expressions, gaze (eye tracking), and body postures that are frequently impaired in ASD. In fact, studies have assessed macaques using the Monkey Social Responsiveness Scale to determine the naturally occurring variation in autistic traits within populations (reviewed in [225]). Kikuchi et al. utilized macaque genotype and phenotype resources to assess the variations responsible for ASD and compared them with clinical data sets of various NDDs from DisGeNET. Macaques showed similar mutations and constraints in genes responsible for NDDs, suggesting they can be used as bonafide ASD models [229].

*Shank3* KO exhibits increased penetrance in cynomolgus monkeys compared to multiple mouse models, and the KO monkeys match the human condition more closely than mice [230] (reviewed in [194]). They are smaller in size and display a range of ASD-relevant phenotypes, including delayed vocalization, anxiety (reduced exploration time in a novel cage, increased duration and frequency of crook-tail, a posture suggestive of fearfulness in juvenile monkeys), stereotypy, reduced eye contact, and decreased glucose metabolism in different brain regions as assessed with PET, many of which could be rescued with fluoxetine, an SSRI [224]. This model also exhibits altered brain connectivity patterns and sleep disturbances [231]. Similarly, TALEN-edited macaques with LoF of the *Mecp2* gene have been developed as an NHP Rett model [192]. Just like RTT patients, males are embryonic lethal, while females exhibit clinical manifestations and have similar dysregulation of immune genes in their blood transcriptome. This points to the stark discrepancy in phenotypes across different animal models of the same disease. Multiple RTT mouse models have been shown to present a phenotype in adult males, while RTT male patients mostly die *in utero* or are born with severe encephalopathy and do not survive beyond two years of age.

Environmental causes of ASD, such as VPA treatment and maternal immune activation, have also been modeled in marmosets [191,232]. Offspring of VPA-treated NHP females were shown to have a reduced number of mature neurons in the PFC and disrupted expression of genes related to neurodevelopment [168].

Notwithstanding their immense value in ASD research, prohibitive maintenance costs and ethical considerations limit the use of NHPs. Additionally, their small sample size precludes studying the heterogeneity in ASD presentation. For example, only 3 successful pregnancies were obtained out of 116 embryos injected with CRISPR guides, and only 1 was a live birth [230]. Due to the long breeding cycles of NHP and 1–2 offspring per pregnancy, it gets extremely challenging to segregate the mosaic variability introduced through CRISPR editing by germline transmission.

## 5. Human ASD Models

Since ASD diagnosis is primarily behavioral, animal models of autism with appropriate face validity have been seminal in deciphering the underlying molecular mechanism. But lab animals are largely inbred as opposed to the outbred human population, and more important, given their evolutionary separation from humans, neuropsychiatric disorders such as ASD cannot be entirely modeled in animals. The human brain is grossly different from the mouse brain, be it in the developmental programs that sculpt the neocortex, underlying transcriptional paradigms, or even the biology of neurons. Human neurons display neoteny, or protracted development, even when transplanted in a mouse brain [233]. Such limitations exacerbate issues with translatability and drug development. This has prompted the use of neural progenitor cells (NPCs) and induced neurons (iNs) derived from human embryonic stem cells (hESCs) and induced pluripotent stem cells (iPSCs). iPSCs represent an excellent tool for studying ASD genetics (https://www.sfari.org/resource/ips-cells/ accessed on 12 June 2025) as iPSC-derived models have been shown to recapitulate several transcriptional and epigenetic paradigms of in utero brain development [234]. Additionally, because iPSCs are derived from a patient’s own cells through reprogramming [235], these cells exactly match their genetic background and provide an invaluable preclinical platform for testing the effect of an intervention (such as a drug) on the patient. Clones derived from congenic iPSCs—some with the gene of interest knocked out using technologies such as CRISPR—can be used to causally attribute genetic contribution to certain molecular events such as cell polarity and synaptogenesis, as well as discriminate causal variants from bystanders. Patient-derived iPSCs can also be used to stratify the heterogeneous ASD population [236].

iPSCs provide a means to study neurogenesis in vitro and serve as a source of iNs, a tissue that is difficult to obtain, especially at ASD-relevant timepoints. Today, iNs are routinely used in high-throughput drug screens, in deciphering circuit-level dysfunctions (reviewed in [36,237,238], protein network convergence [239], as well as in diagnosis. RNA sequencing done on fibroblasts (from 71 individuals of the Undiagnosed Diseases Network) differentiated into iNs increased the diagnostic yield to 25.4%, which was much more informative than sequencing the patient fibroblasts directly [240].

iNs are generated from hPSCs using both directed differentiation as well as overexpression of transcription factors such as neurogenin 2 (*NEUROG2* or Ngn2). hPSCs can either be CRISPR edited or lentivirally infected with inducible Ngn2 to generate iNs that gain immunoreactivity for neuronal markers like MAP2 and TUJ1 in just two weeks [241]. While Ngn2-iNs can be reliably used as a model of excitatory neurons, they are a heterogeneous cell population with wide variations between batches and lines. This has prompted the use of various signaling cues, in addition to Ngn2, that can prime the differentiating hPSCs to a particular regional identity, such as the central nervous system [242]. Additionally, iNs can be profiled using single-nucleus RNA sequencing and benchmarked against gene panels (such as MS-117) to discern their maturation state and compare them to in vivo gene expression in the fetal brain [243,244]. Over the last decade, there has been tremendous momentum in this area of research [245]. iNs can now be maintained for as long as 150 days in vitro in a monolayer culture, and high-throughput protocols to generate LoF alleles to study the breadth of ASD candidate genes have been developed [246].

iNs in 2D culture represent a simplistic, reductive model because these cannot replicate cell–cell and cell–ECM interactions that occur in a 3D tissue environment. The need for more complex 3D models led to the use of brain organoids, which are self-organizing entities derived from iNs under specific signaling cues [247,248,249]. Organoids can model generalized features such as brain size differences, corroborating macrocephaly and microcephaly, variation in the size of specific brain regions such as the cerebellum and ventricular cavities, as well as advanced features such as evolution of network properties like oscillatory waves resembling preterm human EEGs [250]. A classic example of this is the gene dosage-dependent size differences in unguided neural organoids (UNOs) for *PTEN* variants (inversely proportional to *PTEN* expression) [251]. On the other hand, guided neural organoids (GNOs), also known as brain region-specific organoids, are useful for deciphering circuital miswiring in the developing cortex. GNOs like cortical organoids have been shown to generate synthetic spontaneous neural oscillations [250]. 3D organoids spatiotemporally model key neurodevelopmental events and allow access to ASD-pertinent windows for therapeutic intervention, as they can be followed up longitudinally [249,252]. This represents a huge leap over the use of human postmortem tissue, which is often degraded and offers a less relevant timepoint, or low-resolution brain imaging without longitudinal follow up. GNOs have been benchmarked against specific developmental milestones of the human brain using bulk RNA transcriptomic signatures. For instance, 3-month-old cortical GNO models 19–24 gestational weeks to 6 months postnatal brain [253]. Moreover, tools like CRISPR iTracer allow lineage tracing of cells over time as the organoid continues to develop. hPSC-derived organoids have also enabled us to test out epidemiological observations pertaining to the environmental risk factors of ASD, such as maternal infection during pregnancy. Zika virus-infected human GNOs exhibit a reduced growth rate and recapitulate microcephaly [254]. Similarly, the impact of alcohol, nicotine, valproic acid, glucocorticoid, and hypoxia on the development of ASD has also been studied using GNOs [36].

Notwithstanding the wealth of information that can be gleaned from organoids, they can only model a part of an organ at best. However, the complexity that arises through the crosstalk between different cell types is missing in organoids. To address this, assembloids have been developed [255,256]. Akin to organoids, assembloids are stem cell-derived 3D models with intrinsic self-organizing ability, but in addition, they also contain multiple cell types, achieved through fusion of two or more organoids or populating an organoid with a cell type it originally lacks. Assembloids can model circuits better, such as in a cortico-striatal or cortico-thalamic assembloid, reproduce anteroposterior and dorso-ventral polarity when Sonic hedgehog-expressing iPSCs are integrated into a forebrain organoid, and even model neuronal migration like those of interneurons from a ventral forebrain organoid into dorsal circuits when the two are fused to form an assembloid [255]. Assembloids can model emergent “system” properties that the individual component organoids lack. For example, cortico-ventral forebrain assembloids of Timothy syndrome exhibit disrupted GABAergic interneuron migration due to abnormal calcium signaling (caused by mutation in LTCC, a calcium channel Ca_v_1.2) and can be pharmacologically rescued [37,257].

Despite the remarkable progress made in the last two decades, human stem cell-based models still fall short of being an ideal ASD model, primarily because the molecular changes observed here cannot be correlated to behavior. The emergent network-level phenotypes observed in assembloids are still very basic compared to the complex behavioral repertoire of an animal model. Another limitation of these models is the dosage and timing of signaling cues that are ectopically administered and acute in nature, as opposed to sustained, controlled endemic cues that developing brain cells are exposed to.

## 6. A Comparative Analysis of Current ASD Models

To this day, a perfect ASD model does not exist. This is due to several factors. First, there are gaps in our knowledge about how neurons migrate and connect with other neurons to form circuits. Second, circuit-level properties of neurons significantly differ from their individual cellular properties, and these contribute to behavior. Third, ASD diagnosis is based on a set of observed and reported behaviors, which are difficult to model even in NHPs, as correlates of mental states cannot be reliably ascertained across species. At a granular level, the cytoarchitecture of the human brain, in important areas such as the PFC and Brodmann area, differs even from other hominoid species [258].

An ideal ASD model should be human-derived and allow study of emergent network properties in neural circuits, leading to behavioral phenotypes. Until we get to the point where such a model is tenable, we should evaluate ASD models on a 3 × 3 matrix of biological validity, complexity, and scalability (reviewed in [259]). Biological validity is best viewed through the lens of evolution, looking at how far back in time our common ancestor had split. Humans diverged from mice, macaques and chimpanzees ~80, 25, and 6 million years ago, respectively, suggesting that in terms of broad genetic similarity, chimpanzees will be better models for human disease than mice and monkeys. Of course, patient-derived hPSCs share 100% genetic similarity with humans, as do ex vivo human models such as surgically excised or postmortem brain tissue. In terms of complexity, 2D hPSC cultures rank the lowest and NHPs the highest. Although assembloids have been able to recapitulate complex recursive loop circuits [260], they are still in no way close to a real brain, where neural circuits have developed in the microenvironment of spatiotemporal gradients of relevant biological cues. In terms of scalability, 2D hPSC cultures and organoids do much better, especially with recent advancements in automation and robotics that allow large-scale perturbation of multiple ASD gene targets using high-throughput methods such as single-cell and single-nucleus RNA sequencing, BONCAT, spatial transcriptomics, and massively parallel reporter assays [261,262,263,264]. Nonmammalian models such as worms and flies, while scoring very low on biological validity, have been seminal in deciphering the mechanism of action of multiple NDD genes. While *C. elegans* has a short 3-day life cycle, transparent embryos, and a complete connectome map for its 302 neurons [265], most LoF DNVs have been functionally annotated in *Drosophila* through complementation/overexpression-based rescue because of the wealth of genetic tools available for flies [266]. On the contrary, postmortem human brain tissue, while clinically extremely valuable, is confounded by changes that accumulate due to living with ASD and may not be representative of developmental disruptions. In addition, the more advanced the model, the greater is the cost associated with its maintenance. Lastly, both NHP models and clinical samples are inherently limited in number and have stringent regulations for their ethical use.

Multiple ASD animal models have been developed, each with its unique set of opportunities and challenges (reviewed in [225]). One approach to refine model selection would be to apply the criteria of construct and face validity contextually. For instance, abnormal mitochondrial morphology is a conserved phenotype across multiple models of *FMR1* LoF (RNAi in ex vivo human fetal brain tissue, FXS cortical organoids, *Fmr1* KO mice, *Fmr1* knockdown macaque), but there is limited overlap in FMRP brain interactome across species [259]. The co-expression pattern of nitric oxide synthase 1 and FMRP varies between mice and human mid-fetal layer V pyramidal neurons, suggesting that this aspect of ASD cannot be modeled in *Fmr1* KO mice [267]. On the other hand, mouse models have been shown to recapitulate human molecular phenotypes, including protein interactions and alterations in dendritic spine density, for several other ASD risk genes. For example, mice with trisomy 21 phenocopy cell-autonomous oligodendrocyte maturation defects observed in Down syndrome patients [268]. Even the strain of mice used is an important experimental variable here. The effect of genetic variants can differ among backgrounds, as shown for different strains of mice exhibiting varying levels of resilience to loss of one *Chd8* allele [269].

Not only should construct validity be extended to a greater molecular depth as opposed to just modeling the genetic mutation, but also it should be informed by the limitations of each model. Rat models of RTT exhibit impaired juvenile play and regression of acquired psychomotor skills, traits that cannot be modeled in mice [270]. Multiple mouse models (germline line or brain-specific postnatal KO) do not recapitulate the brain development aspect of RTT, as there is a delay in the onset of symptoms (~5–6 weeks old) [115,271], suggesting that *Mecp2* plays a role in the maintenance of brain function in mature neurons in mice. Symptoms appear even later in heterozygous females. In rats, the symptoms appear earlier, with male hemizygous rats being symptomatic by 3 weeks and dying by 2 months [272]. Heterozygous rats are also significantly heavier than littermate WT but have decreased brain weight. Although rat models of RTT demonstrate better construct validity than mice, they are still worse than macaques because *Mecp2* KO cynomolgus monkeys exhibit embryonic lethality of males, just like RTT patients [192,273]. Hence, both the choice of the model and the inferences drawn from studies using a particular model need to be calibrated accordingly.

## 7. Conclusions

Decades of preclinical research in multiple models have illuminated various aspects of ASD. Today, multiple models exist in which the effect of genes and environmental risk factors, separately as well as synergistically, can be studied (Figure 1). The access to hPSC-derived cell types from a patient’s own cells has been a significant milestone in the development of personalized treatments for NDDs and NPDs. Building on this, recent discoveries, such as a cargo of specific non-coding RNAs carried by placental trophoblast-derived exosomes into neural progenitor cells, have opened new vistas in ASD etiology [274]. While animal models may not model the full spectrum of prenatal, perinatal, or postnatal ASD-related behavior or display species-specific differences in drug sensitivity (for example, the anticonvulsant VPA is required in a much lower dose to induce ASD-related phenotypes in pigs (50 to 75 mg/kg) compared to monkeys [149,150]), ASD remains a primarily behavioral disorder that needs to be modeled in live animals. Efforts to determine points of convergence as well as choosing models that can appropriately answer one’s research question remain essential, but increasingly, the community needs to move towards orthogonal validation of findings from a particular model system in other models. hPSC-derived human 2D and 3D ASD models are now routinely being used in preclinical research, and their increasing complexity to simulate the *in vivo* environment more closely holds great promise for the future [275,276].

## Figures and Tables

**Figure 1 cells-14-00908-f001:**
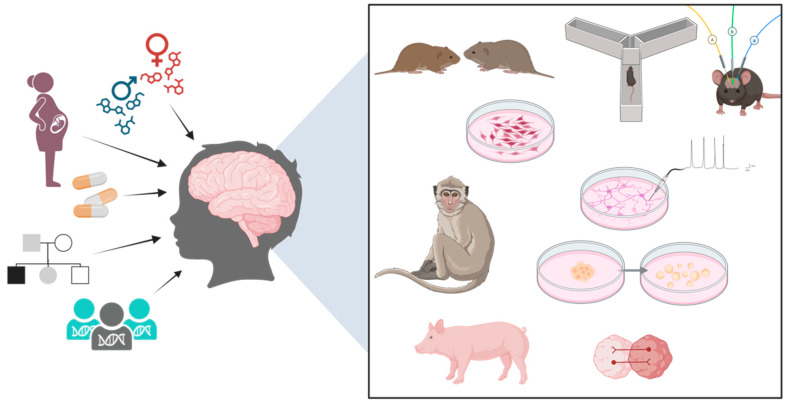
The risk factors for ASD include both genes (rare and common variants) and environment (drugs, hormones, etc.), especially operating while the fetal brain develops *in utero* and extending until early childhood as the human brain continues to develop. Modules of this developmental paradigm have been modeled in rodents, pigs, and monkeys using various behavioral tests as well as in hPSC-derived 2D and 3D cultures using electrophysiology and other functional readouts. Created in https://BioRender.com.

## Data Availability

Not applicable.

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
