# Peer review of "The Evolving Landscape of Functional Models of Autism Spectrum Disorder"

_cells, 2025, doi:10.3390/cells14120908_

Round 1

Reviewer 1 Report

Comments and Suggestions for Authors

Although the article is mostly well-written, it is unclear what this review offers over others in the field, in particular “Translational neuroscience approaches to understanding autism” by Veenstra-VanderWeele et al. 2023. In my opinion, the authors need to present a case for what this review offers that is novel.

However, I like what the authors present when they talk about the use of in vitro systems derived from humans, but by the time I got to that part of the article, it kind of felt like the authors had a hidden agenda to say that this using these in vitro models is the best way forward, rather than their original motivation of comparing models (as described in the abstract). It may perhaps feel this way because the differences described between humans and mice feels overstated, or at least, the similarities between mice and humans are rather understated. There are considerable similarities between brain structure and function across the species that make studies using mouse models appropriate. This is just to say that the authors seem to minimize the use of animal models, and I don’t think that is their intent. Rather, I think the goal is to compare what can be gained from each model and I’m not sure they fully accomplish this goal.

Below are some specific comments on the manuscript:

  • In the introduction, the authors state that “ there are several roadblocks in research towards finding a cure for ASD.” This is a sticky concept, in my opinion. Because people with ASD present symptoms on a wide spectrum, as noted by the authors, there are people on the spectrum who, as the authors note, do not need support and can function independently, and may refer to themselves as neurodiverse. In these cases, there is a portion of the autism population who does not feel like there needs to be a “cure.” It would be beneficial for the authors to clarify what a “cure” is and to perhaps indicate that finding underlying causes for severe neurodevelopmental disorders that may reduce intellectual disability or other disabilities is the goal.
  • In the introduction, the authors note that the effects of the external factors are “maximal in utero, particularly during the critical window of brain development.”
    What is this critical window? Brain development is quite protracted, even through adolescence, making this statement too broad and not very useful.
  • Under 3a, the authors more appropriately refer to “critical windows of development.” But here they could also perhaps be more specific as to when this matters for antidepressant exposure. More importantly and they need to provide some references for SSRIs leading to increased risk of developing ASD.
  • It is surprising/interesting/confusing that the authors describe genetic mouse models for genes that are not included in their list of high confidence ASD genes. TSC1/2, for example. The authors could provide an explanation for including this model in their discussion. They could also provide what the cutoff is for the list of high confidence genes they included in their table. But again, it is unclear what this offers the field, as these genes are probably listed in many review articles about etiologies of ASD.
  • Line 378- Fibroblasts from 71 individuals from the Undiagnosed Diseases Network when transdifferentiated into iNs and subjected to RNA sequencing, increased the diagnostic yield to 25.4%, well beyond patient fibroblasts alone [207]. I don’t understand what the authors are trying to say here.
  • The figure is pretty, but it isn’t clear what it adds to the review article.
  • The section on primates as models is so short, I would question why it is included. What are we learning from this section? That primates are very similar to humans but expensive?

Minor comment:

  • Line 295, what is REST? Can the authors say something like “The transcription factor REST” for clarity?

Reviewer 2 Report

Comments and Suggestions for Authors

Ranjan and Bhattacharya have provided with a very nice and clear manuscript. Very informative and pleasant to read. Perhaps a bit too much focused on SHANK3, leaving aside other important ASD genes as TCF4, ZEB2, MECP2 and others. 
The review is fine like this. However I think it would definitely benefit of some more information on the role of epigenetic modifications in ASD (see for example works on how the chromatin architecture and variations in non-coding DNA influence the transcriptional regulation and how these are linked to ASD and ID). 

Author Response

Ranjan and Bhattacharya have provided with a very nice and clear manuscript. Very informative and pleasant to read. Perhaps a bit too much focused on SHANK3, leaving aside other important ASD genes as TCF4, ZEB2, MECP2 and others. 
The review is fine like this. However I think it would definitely benefit of some more information on the role of epigenetic modifications in ASD (see for example works on how the chromatin architecture and variations in non-coding DNA influence the transcriptional regulation and how these are linked to ASD and ID).

We thank the reviewer for their critique.

We have added some information on TCF4, ZEB2, MECP2 as well as a paragraph on the role of epigenetic modifications in ASD in the revised version of the review article.

Reviewer 3 Report

Comments and Suggestions for Authors

The manuscript titled “The evolving landscape of functional models of autism spectrum disorder” by Jai Ranjan and Aniket Bhattacharya presents a broad and up-to-date overview of model systems used in ASD research, ranging from traditional animal models to advanced patient-derived systems. The authors do a commendable job covering the genetic, environmental, and cellular modeling strategies used in the field.

However, the overall structure would benefit from greater conceptual cohesion. At present, the manuscript reads more like a catalog of models than a narrative-driven synthesis. The review would be strengthened by framing the discussion around key research questions in autism—such as mechanistic insight, translational relevance, or model fidelity—and evaluating how different systems address these. This approach could help readers orient themselves conceptually rather than moving linearly through model types.

The manuscript would also benefit from a more critical comparative analysis of the models. For example, while human iPSC-based models and organoids are rightly emphasized as promising platforms, their technical limitations—such as immaturity, variability, and lack of behavioral correlates—are not fully discussed. Providing a balanced view of both strengths and shortcomings would improve the review’s utility to the reader.

Importantly, the translational relevance of these models is only briefly touched upon. Highlighting examples where preclinical models have successfully led to therapeutic insights (or failed to do so) would significantly enhance the manuscript's impact. Furthermore, while the authors appropriately reference assembloids and advanced stem cell models, additional emerging tools such as BONCAT, spatial transcriptomics, or in vivo imaging could be acknowledged to provide a fuller picture of the field’s direction.

The issue of behavioral readouts in animal models should also be addressed more critically. References to “ASD-like behaviors” in rodents should include acknowledgment of the limited correspondence between mouse behavior and core human ASD symptoms. Greater discussion of construct and face validity would clarify these distinctions.

Better background on high-risk genes like Shank3 and CHD8 (among others) should be given, better reflecting what is known about their role in ASD.

In terms of writing, the manuscript would benefit from some language tightening. There are several instances of repetitive or overly wordy phrasing. Additionally, technical terms such as "idiopathic ASD" and “construct validity” may require brief definitions for readers outside the immediate field.

Table 1 provides helpful detail but could be improved by re-organizing content—either alphabetically or by functional pathways—and by adding a column that briefly summarizes the limitations or unique strengths of each model.

While references are generally strong, the inclusion of a few more recent large-scale or systems-level studies could round out the review. Finally, Figure 1 is referenced but not included in the version reviewed. It will be important to ensure that all figures are present and that they contribute clear conceptual value beyond reiterating the text.

In summary, this is a comprehensive and timely review with the potential to become a valuable resource. To reach that potential, it requires revision with more critical synthesis, improved structure, and greater emphasis on translational relevance. These additions would help the manuscript better serve its intended audience of clinicians, neuroscientists, and translational researchers.

Round 2

Reviewer 1 Report

Comments and Suggestions for Authors

Overall, I like the responses that the authors had to my comments. It appears that the review offers new information for the field. I still feel as though the authors present a bias toward using the human derived model systems. See my comment below. This is a minor point for me, but I do think the tone of the paper would be dramatically altered if the authors consider addressing this comment, which I think would require minimal effort.

Most of my other comments are minor and I do think the paper is well-written. I have only one Major Concern:

Lines 194-197, the authors talk about the transcriptional changes induced in the fetal brain induced by exposure to valproic acid. These changes were observed in the E12.5 mouse. The next sentence seems to compare the E12.5 mouse to the third trimester in humans. E12.5 is mid gestation in mice and has typically been compared to early second trimester in humans. The placenta has only formed at 8.5-10.5, entorhinal cortex neurons start to be born at E13.5 and hippocampal neurons a few days later. Even the paper the authors cite about synaptogenesis is a study performed in postnatal day 4-10 rats, a time period that the cited paper, and the field in general, considers to be similar to the third trimester in humans.

This portion of the added text therefore feels inaccurate and misleading.

Minor concerns:

I appreciate that the authors have worked to improve the manuscript and tone down the feeling that they are pushing the human iPSCs, organoids, and assembloids as the best model to study. I think that perhaps where this falls short is that the limitations of working with the human cells has not been provided, or at least not in as much detail as the other model systems. Therefore, the feeling is still that the authors have a preference for these model systems and it would be more unbiased if they would comment perhaps on the lack of behavioral readouts from these systems, or if the cost is high, or you need specialized equipment, as they have done for other studies. The authors have attempted to do this in the text they added in section 6, but they still feel less critical of these in vitro models because for all other models they have pointed out the limitations throughout the text. In section 5, there is no critique of these models. Perhaps this does not need to be addressed, but the authors should be aware that this is the tone they are setting.

In the last paragraph of section 6, the Rett discussion ends quite abruptly and it ends by saying that rats are better than mice and worse than macaques. This could be worded in a less biased or softened way by pointing out that they better recapulate construct validity than mice, but do not do as well as macaques. This is, again, my opinion about how the paper should be written to provide the impression that the authors are unbiased, even though everyone has biases.

Specific Concerns (Minor)

Lines 188-190: “The effect is maximal in utero, particularly during the critical window of brain development…” It still feels like this is vague. What are the authors considering the critical window of brain development? When in gestation is this occurring and what is occurring?

Lines 213-214. The authors state that the maternal gut microbiome “can” lead to ASD. My personal opinion is that they should say “could” lead to ASD, because “can” indicates possibility but is almost like using the word “does”, whereas “could” shows that this is a possibility with less certainty. The references the authors cite seem like reviews that link maternal gut microbiome with having children with ASD, and I admit I didn’t read these reviews, but I suspect these reviews reveal a relationship between maternal gut microbiome and offspring with ASD but can’t completely indicate causality.

Line 337: These regions are rich in ASD risk genes and are downregulated ASD cases [213]. This newly added text is a bit confusing. Are the authors saying that these regions are downregulated in ASD cases? Is there perhaps a typo here and it should say “downregulated in ASD cases?”

I really like what has been added to the NHP section. However, the portion about VPA exposure now feels abrupt and does not flow well following the Rett discussion. Perhaps the VPA should be its own short paragraph. The last paragraph also could be more clearly written by saying “due to the long breeding cycles of NHP” instead of saying “their.” Changing this wording isn’t a must, but stylistically offers clarity.

Line 525 “observed and reported behavior” should be “behaviors”

Comments on the Quality of English Language

The text could use some light editing, as it is missing prepositions (e.g., line 39 “affecting around 1-3% individuals globally” should be “of individuals”) and articles (e.g., lines 342-351: “In the Autism Mouse Connectome study…” and “…with a significant level of heterogeneity.”). Or at least the manuscript should be double checked to ensure that the prepositions or articles have intentionally been left out.
